# Does Forest Therapy Have Physio-Psychological Benefits? A Systematic Review and Meta-Analysis of Randomized Controlled Trials

**DOI:** 10.3390/ijerph191710512

**Published:** 2022-08-24

**Authors:** Yunjeong Yi, Eunju Seo, Jiyeon An

**Affiliations:** 1Department of Nursing, Kyung-In Women’s University, 63 Gyeyangsan-ro, Gyeyang-gu, Incheon 21041, Korea; 2Department of Nursing, Pai-Chai University, 155-40 Baejae-ro, Seo-gu, Daejeon 35345, Korea

**Keywords:** forest therapy, forest bathing, depression, blood pressure, systematic review

## Abstract

Forest therapy involves visiting forests or conducting forest-based treatment activities to improve one’s health. Studies have investigated the health benefits of forests, but consensus has not been reached. This study comprised a systematic review and meta-analysis to determine how forest therapy affects the physiological and psychological health of adults. The Cumulative Index to Nursing and Allied Health Literature, Embase, and Medline databases were searched on 31 August 2021. Systematic review and meta-analysis, risk evaluation, GRADE evaluation, and advertisement effect evaluation were performed for each article. The effect size was calculated by dividing blood pressure as a physiological indicator and depression as a psychological indicator. Of the 16,980 retrieved studies, 17 were selected based on the inclusion criteria. Of these, eight studies were included in the meta-analysis. The effect size of forest therapy on improving systolic and diastolic blood pressure was not significant; however, it significantly reduced depression. While the results have limited generalizability due to the inclusion of few studies, the effects of forest therapy on reducing depression have been confirmed. Since the application of forest therapy was heterogeneous in these studies, a moderator effect analysis or subgroup analysis in meta-analysis should be performed in the future.

## 1. Introduction

As human life expectancy increases, people are becoming increasingly interested in health promotion. The prevalence of chronic diseases caused by unhealthy behavior has increased [1], psychological stress has increased due to urbanization [2], and people are more interested in living healthily than living long. As a result, people seek out various health-promoting activities; they try to eat healthy food, exercise regularly, reduce stress, expand social relationships, and improve their mental wellbeing. In this regard, forest therapy has received public attention [3].

Forest therapy (or forest bathing) is defined as visiting forests to conduct treatment activities to improve one’s health in a forest environment; it is known to increase immunity and improve health by utilizing various elements of nature, such as fragrance and landscape [4,5].

The main components of forest therapy are walking, experiencing the forest with the five senses (seeing, hearing, touching, smelling, and tasting), forest viewing, forest meditation, Qi-Qong, aromatherapy, herbal tea therapy, and making crafts using natural materials [6]. However, forest activities are focused mainly on walking in the forest, yoga, appreciating the scenery, meditation (such as breathing or walking meditation) and smelling.

Forest therapy incorporates various healing elements, such as sunlight, landscape, temperature, phytoncide, food, sound, and humidity. By enjoying them, humans feel comfortable, their immunity increases, and their health improves. Forest therapy includes not only the healing factor of the forest environment but also the positive experiences of natural scenery and enhances attitudes toward nature [7]. The stress recovery theory and attention recovery theory are representative theories of the mechanism by which forest therapy provides healing to humans. Ulrich et al. established the stress recovery theory, stating that the natural environment not only brings changes to one’s physiological indicators—such as the human heartbeat cycle, muscle tone, skin conductance, and blood pressure (BP)—but also improves one’s emotional state [8]. Kaplan established, in the attention restoration theory, that exposure to the natural environment means being away from daily life, which reduces fatigue because no effort or concentration is required [9].

Several studies have revealed the physiological and psychological health effects of forests. Physiologically, the effect has been confirmed through the improvement of BP [10,11], heart rate (HR) or heart rate variability (HRV) [12,13,14], cortisol levels [15], pain relief [16], and respiratory function [17,18]. Psychologically, it has been confirmed through its effects on depression [10,17], anxiety [18,19,20], quality of life (QOL) [11,21], mood [13,14,22], and emotional burnout [23]; and cognitively, on concentration [24] and cognitive function improvement [25]. In addition, it was said that the body’s immune function and the activity of anti-cancer proteins (such as perforin, granulysin, and granzyme A/B-expressing cells in intracellular) were enhanced, and this enhanced immunity lasted for one month [26]. Studies have shown that biogenic volatile compounds in forests improve human immunity to prevent infectious diseases, such as the coronavirus disease [27], and even reduce mortality from the same [28].

However, previous studies only conducted a systematic literature review on the physiological or psychological effects of forest therapy [29,30], only targeted the patient group [31], and did not conduct a meta-analysis [29]. While a previous meta-analytical study [32] did confirm that forest therapy is effective in reducing depression, there was no detailed analysis of specific forest healing activities, and the sample was highly heterogeneous as it included both children and adults.

Therefore, this study sought to specifically analyze the effectiveness of forest therapy through a systematic literature review and meta-analysis for both physiological and psychological areas, respectively.

Through this study, we intend to find out whether forest healing is effective in promoting the health of adults. In addition, we would like to address various other aspects by looking at the side effects of forest healing activities.

## 2. Materials and Methods

### 2.1. Research Design

This study comprises a systematic literature review and meta-analysis designed to integrate and analyze the results of randomized experimental studies that confirmed the physiological effects and the psychological health effects of forest therapy. This study was conducted in accordance with the systematic literature review handbook of the Cochrane Coalition and the systematic literature review reporting guidelines of the Preferred Reporting Items for Systematic Reviews and Meta-Analyses (PRISMA).

### 2.2. Literature Search Strategy

Based on the PICOS (Population, Intervention, Comparison, Outcomes, and Study) framework, the research questions for a systematic literature review are as follows: This study targeted adults aged 19 or older who had previously participated in forest therapy. The measurement tools for children were different from those for adults, so children were not included. Additionally, children were excluded from our study to reduce heterogeneity while performing the meta-analysis. The intervention included using forest therapy. Forest therapy includes walking in the forest, sensory experiences (seeing, hearing, smelling, etc.), performing meditation or yoga, and group activities. The comparison included cases of walking or resting outside the forest, or without any intervention. The study outcomes included indicators of physiological responses and indicators of psychological responses. This study did not limit the outcome variables to the effects of forest therapy, because the purpose of this study was to explore the indicators of various types of physiological and psychological responses and to verify their effects. Only randomized experimental studies were included in this research; clinical studies, observational studies, and case studies without randomization were excluded.

The period of publication of papers to be searched and analyzed was from 2011 to 31 August 2021. The period was limited to the last decade in the interest of understanding recent trends, and we predicted that high-quality studies with randomized controlled trial methods would be found within this period. The Cumulative Index to Nursing and Allied Health Literature (CINAHL), Embase, and Medline databases were searched. In addition, a hand search was conducted using the citations of the research papers and search terms from Google Scholar. Medical Subject Headings (MeSH) terms and Emtrée were checked using a systematic search, and the following terms were used as a result: “forests”, “therapy”, “baths”, “shinrin-yoku”, “nature”, “bathing”, “healing”, “intervention” and “program”. The search formula was [(forest) and (therapy or baths)] or [(forest or shinrin-yoku or nature) and (bathing, healing, intervention, or program)] (Appendix B).

### 2.3. Literature Selection and Exclusion Criteria

The criteria for selecting literature were: (1) a randomized experimental study on forest therapy with adult subjects (including both healthy and people with diseases), and (2) an academic paper. The exclusion criteria were: (1) studies not published in English by title, (2) studies that were quasi-experimental designs or partially reported, such as presentations at academic conferences, and (3) duplicate literature. The search was conducted in the electronic database according to the search formula, and duplicate papers were removed using the Endnote program. Thereafter, by reviewing the title and abstract, the literature was selected according to the selection and exclusion criteria. To select the final literature, the full text of the literature was examined. In cases where there were disputes between the three researchers, the final literature was decided through discussion. This process was visualized using a PRISMA flowchart (Figure 1).

### 2.4. Quality Evaluation of Literature

Three researchers independently conducted a quality assessment of the literature using the Cochrane risk of bias tool. The seven criteria used in the Cochrane quality evaluation are random sequence generation, allocation concealment, blinding of participants and personnel, blinding of outcome assessment, incomplete outcome data, selective reporting, and other potential biases. There were no other potential biases [33]. The risk of the final selected paper was evaluated using these criteria. If there was a disagreement among the researchers, it was reviewed through discussions and a conclusion was drawn. To assess the quality of evidence, we used the Grading of Recommendations Assessments, Development, and Evaluation (GRADE) method [34]. GRADE has five categories: study design limitations, inconsistencies, indirectness, inaccuracy, and publication bias. Three researchers categorized the evidence’s quality as high, moderate, low, or critically low.

### 2.5. Data Extraction

The characteristics of the final selected paper were analyzed and organized according to the data organization form. The data organization form was divided into physiological indicators and psychological health indicators according to the type of variable. The contents of the data organization form included the author, publication year, subject group, intervention characteristics (intervention time, number, and duration), number of subjects, outcome variables, measurement tools, and intervention results.

### 2.6. Data Analysis

For the statistical analysis of the effect of size integration, homogeneity test, bias assessment, and evidence evaluation, a meta-analysis was performed using the Cochrane Review Manager (RevMan 5.4.1, Seoul, Korea). In the meta-analysis model, the fixed-effects model and random-effects model were mainly used. The final selected studies were analyzed using a random-effects model because heterogeneity was observed in the subjects’ characteristics and the outcome variable measurement tool. In the studies, the effect size was calculated using the standardized mean difference (SMD) because the outcome variable was a continuous variable presented as the mean and standard deviation, and the intervention result was not measured with the same tool. The effectiveness of each described outcome variable and the 95% confidence interval (95% CI) were analyzed using an inverse variance. The heterogeneity between the included studies was visually confirmed through a forest plot and was statistically reviewed using the I^2^ value.

## 3. Results

### 3.1. Selection of Studies

Based on the literature search pursuant to the PICOS framework, a total of 16,980 papers were found. Of these, 735 duplicate papers were excluded, leaving 16,245 papers. Three researchers reviewed the titles and abstracts of these documents, and the criteria for review were: core questions, selection, and exclusion criteria. A total of 16,108 studies were found to be unrelated to the core questions, or the research designs did not meet the selection criteria—resulting in 137 remaining studies. The original text of the remaining studies was then reviewed according to the same criteria and processes.

After excluding 107 studies that were not related to forest therapy, eight studies that were not randomized, and two studies with gray literature (e.g., oral presentation), one study was added through Google Scholar and citation searching. Therefore, a total of 17 studies were finally selected. A systematic literature review was conducted using these 17 studies, and a meta-analysis was conducted on eight studies that were capable of effect size analysis in consideration of the intervention method, outcome variables, and whether treatment was performed in the control group (Figure 1). A meta-analysis was performed as the number of papers available for meta-analysis was met [35]. Among the outcome variables, only blood pressure and depression were available for meta-analysis. A total of eight papers were included in the meta-analysis; six studies measured BP, and four studies measured depression.

### 3.2. Characteristics of Studies in Systematic Literature Review

The characteristics of the 17 studies on forest therapy included in this study are shown in Table 1 and Table 2. The effective outcomes of forest therapy were categorized into physiological health and psychological health and are presented in Table 1.

#### 3.2.1. Participant Characteristics

Among the studies included in the review, nine studies dealt with healthy adults [10,11,12,20,23,36,38,39,40], and eight studies were done with patients [13,14,15,16,21,22,37,41]. Two studies targeted hypertensive patients [13,14]. One study targeted coronary arterial disease [37], chronic obstructive pulmonary disease [15], chronic stroke [21], chronic back pain [22], chronic heart failure [41], and alcoholism [19]. Adults were searched according to the PICOS, and among them, 11 studies targeted only adults [10,11,12,19,20,22,23,36,38,39,40]. There were two studies [21,37] involving both adults and older adults and four studies [13,14,15,41] targeting only older adults. In most studies, the gender of participants was a mixture of men and women, but there were three studies [23,39,40] targeting only men and one study [20] targeting only women.

#### 3.2.2. Intervention Characteristics

Interventions in primary studies were conducted at various times of the year, though this detail was omitted from some reports [12,21,39]. Forest therapy intervention varied from a minimum of 15 min to a maximum of five hours, and the duration of the intervention varied from one day to eight weeks. The number of interventions performed during the intervention period ranged from 1–16, with eight studies with five or fewer times [12,13,20,23,36,38,39,40], four studies with 6–10 times [15,22,37,41], and three studies with more than 10 times [10,11,14]. One study reported that it operated on a three-night, four-day schedule [21].

The intervention methods could largely be divided into two categories: walking or other activities in a forest or city and being outdoors or indoors. Twelve out of seventeen studies involved walking in a forest [10,11,12,14,15,20,21,23,37,38,40,41]. In addition, one study each was conducted for hiking [22], biking [36], camping [17], and viewing [39].

To control for the influence of parameters affecting the surrounding environment and subjects, various controls were used in each study. To maintain the homogeneity of the experimental and control groups, the researchers had participants stay in the same hotel or a hotel with a similar environment [13,14,15,22,23,38,40]; limit physical activity, smoking, alcohol, and caffeine intake [13,14,20,36,37,38,39,40,41], and maintain the treatment conditions of forest and urban environments such as walking speed, course length, and roadside [39,41]. However, some studies did not mention related explanations [10,11,12,17].

#### 3.2.3. Measurement and Outcome Characteristics

BP was divided into systolic BP (SBP) and diastolic BP (DBP). SBP [39] or DBP [12,14,36,37,39] decreased significantly more in the forest therapy group than in the control group. In four of the six studies measuring cortisol levels, they were lower in the forest therapy group than in the control group [12,15,23,39].

Heart rate (HR) also decreased in the forest therapy group compared to the control group [20,39]. In the experimental group (forest walking group), high frequency (HF), which is a component of HRV as a marker of cardiac parasympathetic control, increased [13,20,39,40], and the low frequency/high frequency (LF/HF) ratio, which is a marker of the sympathetic to parasympathetic autonomic balance, decreased [13,20,39,40]. In one study, the forest therapy group had a lower frequency (LF) than the control group [13]. LF is an index of cardiac sympathetic control.

In one study, the forest therapy group reported an increase in saturation of percutaneous oxygen (SPO_2_) compared to the control group [13], but in another study, the comparison between the experimental group and the control group [38] was not statistically significant.

As for the effect of forest therapy on psychological health, among the seven studies that measured mood with the POMS, some studies found that the tension-anxiety subscale [13,15,20,23,40,41] and the anger-hostility subscale decreased [14,20,23,40,41], while others found that the depression subscale [13,14,20,23,40], and the confusion subscales decreased [13,14,20,40,41]. Four studies found that the vigor subscale increased [13,20,23,40]. In three studies measuring anxiety with the Spielberger State-Trait Anxiety Inventory (STAI), the forest therapy group had lower anxiety levels than the control group [20,21,41]. In another study [21], where depression was measured using the Beck Depression Inventory (BDI) and the 17-item version of the Hamilton Depression rating scale (HAM-D17), both indicators showed lower depression levels after forest therapy.

QOL was measured by a QOL scale and a 36-item short-form health survey (SF-36), and the forest walking group and the green exercise and balneotherapy groups had significantly higher scores than the control group [9,22]. Adverse events, such as insect bites, stings, and pollen allergy related to forest therapy, were not reported in any of the studies included in this analysis (Table 1).

### 3.3. Assessment of the Quality of Literature

#### 3.3.1. Risk of Bias

A critical review of the literature was conducted using Cochrane’s risk of bias tool, and it was analyzed through RevMan according to the quality evaluation criteria (Figure 2, Appendix A). Seventeen studies were found to have a low risk of selective outcome reporting and other potential threats to validity. However, 14 studies (82.3%) that did not specify the blinding aspect to participants and researchers showed uncertain risks, as did seven studies that did not report accurate figures or reported only parts of the study results, such as graphs.

#### 3.3.2. GRADE assessment

We used the GRADE profiler program to evaluate the evidence level. The GRADE profiler tool determined that the quality of evidence for forest therapy was “low” or “moderate”. Only two studies were rated “high”.

### 3.4. Estimation of the Effect Size of Forest Therapy

#### 3.4.1. Blood Pressure (BP)

A meta-analysis was conducted by dividing the effect of reducing BP through forest therapy by SBP and DBP. As shown in Figure 3, SBP decreased to 0.24 (*n* = 290, MD = −0.24, 95% CI −2.70 to 2.23) in the forest group, which was not significant (Z = 0.19, *p* = 0.85). The heterogeneity between the two studies was low (Higgins I^2^ = 19%). DBP was found to increase by 0.94 (*n* = 242, MD = 0.94, 95% CI −3.20 to 5.07), which was not significant (Z = 0.44, *p* = 0.66). Heterogeneity was high (Higgins I^2^ = 81%).

#### 3.4.2. Depression

In the meta-analysis, the effect on depression was measured using the BDI [11,12] and POMS [14,20]. As shown in Figure 4, there was a reduced depression effect of 1.46 points in the forest therapy group (*n* = 255, SMD = −1.46, 95%CI −2.80 to −0.12), and the effect size between the forest therapy group and the control group showed a significant difference (Z = 2.14, *p* = 0.03). The heterogeneity was high (Higgins I^2^ = 95%).

## 4. Discussion

This study examined the literature comprising randomized controlled trials conducted on adults to confirm the effect size of forest therapy on health. A total of 17 studies were selected after the search, but only eight studies were meta-analyzed. The discussion tends to refer to the qualitative synthesis of the results of the systematic review and the quantitative synthesis of the results of the meta-analysis. Qualitative comprehensive analysis of selected studies through systematic literature review can serve as basic data for consideration of forest therapy.

The main characteristics of the 17 studies were as follows: First, it was found that the age and gender of the subjects in the forest therapy varied. This was mainly catered to healthy adults; this result shows that the purpose of forest therapy is health promotion rather than disease treatment. When examining the effectiveness of forest healing, whether there is a difference in age or sex is an important consideration. In all 17 studies selected in this study, it was confirmed that age or gender differences were not considered in the analysis. In the study by Kim et al. [42], which investigated individual preferences for forest therapy, women preferred forest therapy in the form of psychology-based treatment, meditation-based treatment, and respiration yoga-based treatment, whereas men preferred forest therapy including mountaineering. However, it was reported that there was no statistical difference in the preferences of men and women for walking and viewing the forest. In the above study [42], in the preference for forest therapy by age, it was found that older adults greatly preferred forest therapy, including plant-based treatment, compared to young adults. Among the selected studies, there were three that included climbing or hiking, both of which were for adults. In forest therapy in the form of climbing or hiking, older adults and women are physically burdened and there is a possibility of injury, so safety management is additionally required. There are differences in the individual needs for forest therapy. Therefore, if a participant has a gender-or-age-specific disease, the contents of the forest therapy program can be individualized to ensure safety considering his/her age and gender. Park et al. [43] conducted an expert Delphi survey on the suitability of forest therapy and recommended it for people with respiratory diseases, stress-related diseases, and somatoform disorders.

Second, the components, duration, and number of sessions were very diverse. While most of the studies involved a single intervention consisting of walking alone, some studies included multiple interventions such as biking, hiking, climbing, games, meditation, etc., in addition to walking. If there are no standard guidelines for forest therapy, Ohe et al. [44] recommended a therapy period of 3 to 5 days for the relaxation effect of forest therapy to continue and lead to stress reduction regardless of the content of the therapy.

Third, forest therapy cannot be done in a double-blind setting because of the movement to a specific location. Due to these limitations, it is very important to maintain the homogeneity of the experimental and control groups and to control exogenous variables for the randomized controlled trials. Of the 17 selected studies, 10 studies mentioned environmental settings for maintaining homogeneity, 16 studies explained measurement protocols, and 11 studies stated how exogenous variables were controlled. To maintain the homogeneity of the experimental and control groups, the participants in both groups stayed in the same hotel, and to control exogenous variables, physical activity, smoking, drinking, and caffeine were restricted during the intervention period. Additionally, to maintain the same walking activity of the experimental group and the control group, walking speed, course length, and roadside were checked. When measuring variables such as physiological indicators, an accurate protocol is important, and especially when estimating the difference between the experimental group and the control group, it is very important to maintain a homogeneous environment setting between the two groups. However, among the selected studies, the protocol for measurement was different or some studies did not provide sufficient explanation for the control of exogenous variables. If the measurement is different, it is difficult to integrate the research results because consistent results cannot be obtained.

In the selected studies, forest therapy was applied in a very diverse program. The program was divided into a one-day and a multiple-day program, but the effect size according to the days of the programs was not consistent. According to previous studies [45,46], the pleasure experienced when interacting with others has a positive effect over fun alone, and interactions with others strengthen an individual’s emotional experience dynamically.

Finally, physiological indicators were measured in a relatively consistent manner, while psychological indicators were measured using various measurement tools. Although physiological indicators showed similar results across studies, psychological indicators were not significant, or statistical values were not presented in some cases. In other words, it was difficult to synthesize the results of the studies selected for systematic review due to their generally non-homogeneous characteristics and because many studies did not provide statistical values for meta-analysis. As a result, only some studies were included in the meta-analysis.

As a result of the meta-analysis, forest therapy was found to be effective in reducing depression in adults (SMD = −1.46, CI: −2.80–0.12). In fact, since the measurement of depression in the primary study included in the meta-analysis is not a measurement based on a doctor’s diagnosis, it is an accurate conclusion to state that forest therapy is effective in reducing the symptoms of depression in adults. A study by Yeon et al. [47], which included a quasi-experimental design in meta-analysis, also reported that forest therapy was effective in reducing depression.

According to Doimo et al., who systematically analyzed both quantitative and qualitative research, forest therapy was found to improve affection, depression, mood, and anxiety, resulting in psychological stability [48]. Stier-Jarmer et al., also conducted a meta-analysis that did not restrict age and found that forest therapy was effective in reducing depression. However, the quality of the studies included in the analysis was low and heterogeneous was very high [49]. Furthermore, in this study, the effect size of forest therapy on depression was very small and the primary studies used for meta-analysis were heterogeneous. Two studies out of the primary studies were found to have a high risk of bias in the quality evaluation. In the GRADE evaluation of the primary studies, two studies [10,20] had a low level of evidence, so it can be judged that evidence for the overall effect size is low. The heterogeneity of the primary studies was likely caused by the variety of measurement tools (e.g., BDI, HAM-D17, POMS), and program types for depression (e.g., walking, sitting, sense experience). The age (e.g., young adults, adults, elderly) and health status (e.g., healthy people, hypertensive patients, patients with chronic stroke) of the participants in the experiment group of the primary studies were different. Since the sessions and duration of the program also varied, it is difficult to determine whether the overall effect size is a short-term effect or a long-term effect. To meta-analyze depression, subgroup analysis is required to identify the cause of heterogeneity, but in this study, subgroup analysis was not possible due to the small number of primary studies. Although it was not possible to identify the cause of heterogeneity through the statistical technique of Meta-ANOVA, the study produced an important finding that forest therapy is effective in reducing depression.

In a systematic review by Mathias et al., forest therapy had physiological and psychological effects; especially among participants with poor psychological health, such as patients with depression, forest therapy had a positive effect as a means of cognitive behavioral therapy [50]. In a systematic review by Wen et al., 11 out of 14 studies reported the effect of reduced depression and forest therapy was effective in relieving negative emotions, including depression [51]. Due to these positive psychological effects, forest therapy has been used as a therapeutic tool for people exposed to psychological stress, such as people suffering from alcoholism, chronically ill people, and patients with pain and insomnia. For this reason, forest therapy is sometimes used as a complementary and alternative therapy to relieve chronic physical symptoms, thereby improving psychological health [48,52]. The effects of forest therapy on health promotion or stress reduction in healthy people are known, with relatively consistent results from previous studies [53].

This study also analyzed the effect size on BP among the physiological effects of forest therapy. The meta-analysis of six primary studies showed that the overall effect size was not significant. Although SBP decreased in the experimental group, the difference was not significant, and the heterogeneity between studies was not high (I^2^ = 19%). In the five primary studies, DBP increased in the experimental group, but it was not significant, and the heterogeneity between studies was very high (I^2^ = 81%). The effect size on BP was not significant because the difference between the experimental group and the control group was not significant even in the primary studies and the variance between studies in the primary studies was large. In previous research [52,54,55,56], BP was verified as a short-term effect of forest therapy; even if there was an effect, the effect size was small, or there were limitations because it was a significant result in a one-group pre-post design.

BP is a parameter for the effectiveness of interventions, but it can be used only limitedly to measure the immediate effects of interventions. In addition, BP is greatly affected by medical conditions such as hypertension and endocrine diseases; it varies according to the circadian rhythm and there are individual differences depending on food habits, such as caffeine intake [57]. In other words, it is a measurement that is difficult to control for confounding variables during the intervention period. In a previous systematic literature review (a total of 28 studies) [58], six studies used BP as an outcome variable to examine the effect of forest therapy, and only three studies indicated statistical significance. The results in which the effect of blood pressure was not significant in this study are due to the inconsistent results of the primary study. BP is an important outcome variable in the study to examine the effects of forest therapy. However, researchers should be aware that there are many factors to consider when measuring BP.

Although the effect of forest therapy showed significant results only for depressive symptoms among psychological outcomes in this study, forest therapy is useful for health promotion. Forest therapy is a low-risk and low-cost intervention that does not threaten people’s health because it entails approaching nature as it is. It can be safely utilized by vulnerable groups (e.g., patients, pregnant women) [59] and can be used as a complex intervention because it can be combined with other interventions, such as walking, meditation, and relaxation therapy [60]. According to the results of the systematic analysis of this study, adverse events of forest therapy were not confirmed, but there is a possibility of adverse effects being caused by environmental characteristics not suitable for certain individuals. For example, insect bites, pollen triggering an allergic reaction, slipping on the road, etc., should be considered during forest therapy. Thus, it is necessary to consider the health status and physical function of the participants.

As shown in this study, it is difficult to clearly explain how forest healing works on depression reduction. The mechanism by which the forest environment affects health can probably be explained by the interaction of forest environmental factors (e.g., phytoncide, oxygen, and forest microclimate) and the five human senses. Based on the theories of Ulrich et al. [8] and Kaplan [9], it is understood that the relaxed environment of nature, away from daily life, brings about a reduction in stress and, consequently, a decrease in depression.

### 4.1. Clinical and Research Implications

Since we confirmed that forest therapy has a positive effect on depression, we would like to emphasize the need for expanding forest therapy to reduce depression in adults. The results of the systematic review in this study can be used as clinical considerations or guidelines on the application of forest therapy. Specifically, natural rather than urban forests are suitable for creating a healing environment, and physio-psychological indicators are affected by circadian rhythms or hormones, so they should be measured based on standard protocols. When forest therapy is applied to patients with diseases, it may be necessary to accompany medical staff to prevent harmful events. To have the effect of stress relief or relaxation through forest therapy, a flat or gently inclined path would be suitable. In addition, to maximize the effect of forest therapy, foods or drinks that can cause sympathetic nerve activation should be restricted. In addition, the season and climate in which the participant will feel comfortable should be considered.

### 4.2. Limitations

This study has certain limitations. First, primary studies were extracted through a systematic search; however, only a few studies were included in the meta-analysis. Because there were few studies used in meta-analysis, subgroup analysis (e.g., gender, absence of disease, number of sessions, program type) and publication bias analysis (e.g., funnel plot) could not be performed to identify the cause of heterogeneity in primary studies. Second, as the systematic literature search did not include all studies published in various languages, it should be considered when interpreting the research results. In this study, the definition of forest therapy was defined as including all activities in the forest. However, since the intensity of activities (e.g., walking) in the primary study was different, the effect of exercise cannot be excluded.

## 5. Conclusions

In this study, forest therapy did not have a significant effect size in decreasing BP, whereas it had a significant effect size in reducing depressive symptoms. Although there is a limitation that the evidence for the result is low, it has research significance because it is the result of a meta-analysis including only RCT studies. The findings of the qualitative and quantitative synthesis of forest therapy performed in the study are expected to present a new therapeutic strategy for depression reduction. Although there is a possibility that the natural elements of the forest may have a harmful effect, if close attention is paid considering the participants’ individual health characteristics, forest therapy will be a good therapeutic program.

## Figures and Tables

**Figure 1 ijerph-19-10512-f001:**
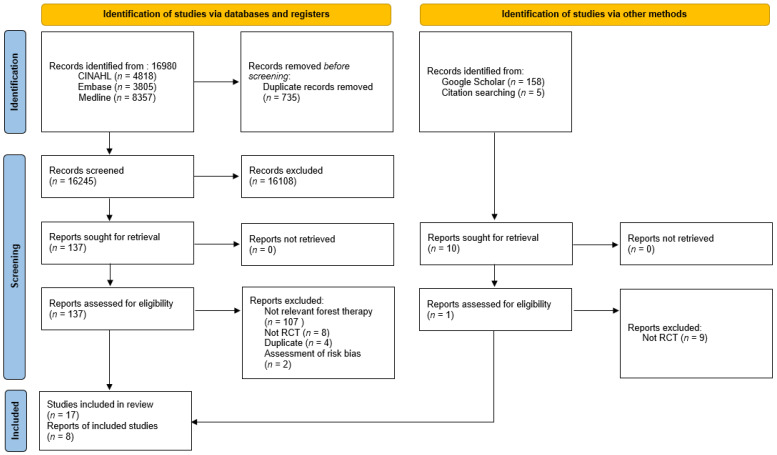
Flowchart of the study selection process.

**Figure 2 ijerph-19-10512-f002:**
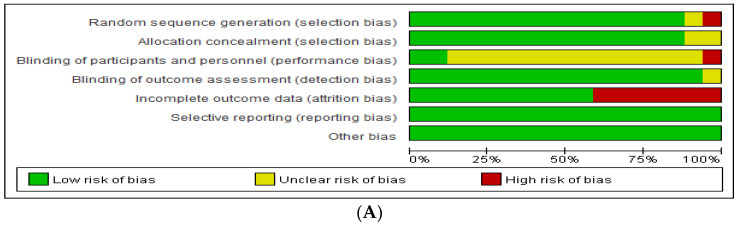
(**A**) Risk of bias graph. (**B**) Risk of bias summary in included studies [10,11,12,13,14,15,17,20,21,22,23,36,37,38,39,40,41].

**Figure 3 ijerph-19-10512-f003:**
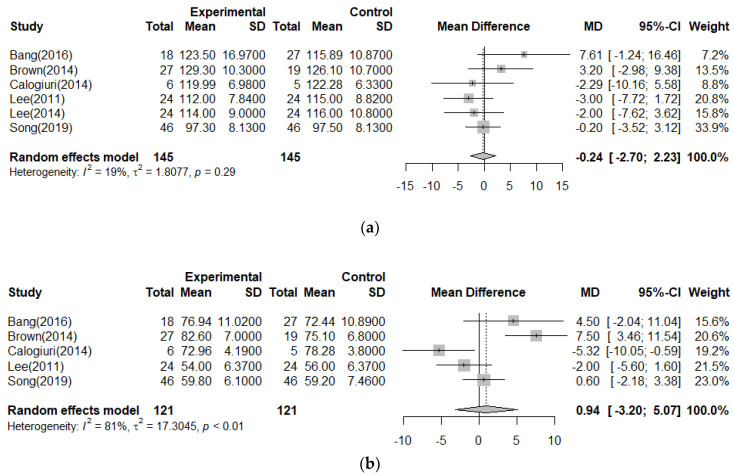
Meta−analysis of blood pressure [10,11,20,36,39,40]. (**a**) <SBP>; (**b**) <DBP>.

**Figure 4 ijerph-19-10512-f004:**
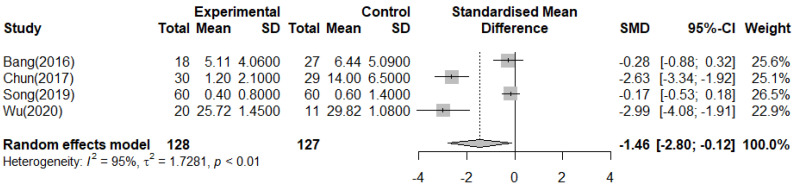
Meta−analysis of Depression [10,13,20,21].

**Table 1 ijerph-19-10512-t001:** General characteristics of included studies.

Category	First Author (Year Published)	Including in Meta-Analysis	Heath Status of Subject	Subjects Size (n)	Age of Subject	Protocol for Measurement	Measures	Main Outcome	Adverse Events	GRADE
Physiological health indicators	[10] Bang (2016)	O	Healthy person	(Exp.) Urban forest walking (*n* = 18) (Con.) No intervention (*n* = 27)	Adults (Male and female)	Promised time	BP	[SBP] Exp. (123.50) and Con. (115.89) were not statistically different [DBP] Exp. (76.94) and Con. (72.44) were not statistically different	NM	⨁⨁◯◯ Low
	[11] Brown (2014)	O	Healthy person	(Exp1.) Urban forest walking (*n* = 27) (Exp. 2) Urban street walking (*n* = 27) (Con.) No intervention (*n* = 19)	Adults (Male and female)	On site (physiological data), online (psychological data) Following the 8-week intervention	HR BP	[SBP] Exp. 1 (129.3), Exp. 2 (129.7), and Con. (126.1) were not statistically different [DBP *] Exp. 1 (82.6), Exp. 2 (80.1) were higher than Con. (75.1) [HR-resting] Exp. 1 (68.3), Exp. 2 (65.1), and Con. (64.0) were not statistically different	NM	⨁⨁⨁◯ Moderate
	[36] Calogiuri (2016)	O	Healthy person	(Exp.) Nature exercise (*n* = 6) (Con.) Indoor exercise (*n* = 5)	Adults (Male and female employee)	After each session, between 8:00 and 9:00 AM (BP, saliva cortisol, blood cortisol) 15:00, 17:00, and 21:00 (blood cortisol)	BP Cortisol (serum)	[SBP] Exp. (119.99) and Con. (122.28) were not statistically different. [DBP *] Exp. (72.96) was lower than Con. (78.28). [Cortisol] Exp. (362.92) and Con. (343.78) were not statistically different.	NM	⨁⨁◯◯ Low
	[37] Grazuleviciene (2016)	X	Patient (CAD)	(Exp.) Pine forest walking (*n* = 10) (Con.) Urban street walking (*n* = 10)	Adults and elderly (45~75 years) (Male and female)	Every day (7 days) 12:00~15:00. Before walking, At 1 min and 60 min after the initial exposure	BP Cortisol (saliva) HR	[DBP *] Exp. (−4.00) was lower than Con. (0.00) (change of baseline between day 1 and day 7) [DBP *] Exp. (−6.00) was lower than Con. (2.00) (change of 60 min after the walking between day 1 and day 7)	NM	⨁⨁⨁◯ Moderate
	[15] Jia (2016)	X	Patient (COPD)	(Exp.) Forest walking (*n* = 10) (Con.) Urban walking (*n* = 8)	Elderly (Male and female)	Before breakfast the day after intervention	Cortisol	[Cortisol *] Exp. was lower than Con. ^†^	NM	⨁⨁◯◯ Low
	[14] Mao(a) (2012)	X	Patient (HTN)	(Exp.) Forest walking (*n* = 12) (Con.) City walking (*n* = 12)	Adult	Measured 30~40 min after intervention (physiological data), next evening (psychological data)	BP HR	[SBP *] Exp. was lower than Con. ^†^ [DBP *] Exp. was lower than Con. ^†^ [HR] Exp. and Con. were not statistically different ^†^	NM	⨁⨁◯◯ Low
	[23] Mao(b) (2012)	X	Healthy person	(Exp.) forest walking (*n* = 10) (Con.) city walking (*n* = 10)	Adult (Male university students)	Before breakfast (on the intervention day and the next day after 2 day intervention)	Cortisol	[Cortisol *] Exp. was lower than Con. ^†^	NM	⨁⨁◯◯ Low
	[12] Niedermerier (2017)	X	Healthy person	Total = 42 (Exp 1.) Outdoor mountain hiking (*n* = NM) (Exp 2.) Indoor walking (*n* = NM) (Con.) No intervention (*n* = NM)	Adult	NM	BP Cortisol HRV	[SBP *] Exp.1 (121.3) and Con. (119.0) were higher than Exp. 2 (119.8) [DBP *] Exp 1. (78.3) and Con. (73.5) were higher than Exp. 2 (72.6) [Cortisol *] Exp. 1 (1.8) and Exp. 2 (1.8) were lower than Con. (2.3) [HRV-LF *] Exp.1 (2967) and Con. (2622) were higher than Exp. 2 (2614) [HRV-HF] Exp.1 (2409), Exp. 2 (1548), and Con. (1581) were not statistically different	NM	⨁⨁⨁◯ Moderate
	[20] Song (2019)	O	Healthy person	(Exp.) forest walking (*n* = 72) (Con.) city walking (*n* = 72)	Adult (Young female university students)	5 min after intervention	BP HRV HR	[HF **] Exp. (105.12) was higher than Con. (57.11) [LF/HF **] Exp. (6.10) was lower than Con. (8.19). [HR **] Exp. (87.0) was lower than Con. (95.6) [SBP] Exp. (97.3) and Con. (97.5) were not statistically different. [DBP] Exp. (59.8) and Con. (59.2 mmHg) were not statistically different. [Pulse *] Exp. (69.3) was lower than Con. (71.9).	NM	⨁⨁◯◯ Low
	[13] Wu (2020)	X	Patient (HTN)	(Exp.) Sit in forest (*n* = 20) (Con.) Sit in suburban (*n* = 11)	Elderly	After intervention	BP HRV SPO2	[SBP] Exp. and Con. were not statistically different ^†^. [DBP *] Exp. (67.95) was lower than Con. (71.64) [HRV-LF *] Exp. (35.0) was lower than Con. (50.88) [HRV-HF *] Exp. (60.54) was higher than Con. (48.37) [HRV-LF/HF *] Exp. (0.68) was lower than Con. (1.36) [SPO2 *] Exp. (98.1) was higher than Con. (97.55)	NM	⨁⨁⨁◯ Moderate
	[38] Zeng (2020)	X	Healthy person	(Exp. 1) Large species of cluster bamboo forest (*n* = 30) (Exp. 2) Bamboo sea site (*n* = 30) (Exp. 3) Bamboo park (*n* = 30) (Con.) Urban (*n* = 30)	Adult	15 min after intervention	BP SPO2	[SPO2 *] Exp. 2 (97.47) after viewing was higher than before viewing (97.37) No statistical comparison between group (Exp. vs. Con.)	NM	⨁⨁◯◯ Low
	[39] Lee (2011)	O	Healthy person	(Exp.) Forest viewing (*n* = 24) (Con.) Urban viewing (*n* = 24)	Adult (Male)	35~90 min after intervention	BP Cortisol HR HRV	[LF/HF **] Exp. was lower than Con. ^†^ [SBP *] Exp. (116) was lower than Con. (118) [DBP] Exp. (54) was lower than Con. (56) [Cortisol] Exp. (0.34) and Con. (0.43) were not statistically different [HR *] Exp. (66.4) was lower than Con. (71.7) [HF **] Exp. was higher than Con. ^†^	NM	⨁⨁⨁⨁ High
	[40] Lee (2014)	O	Healthy person	(Exp.) Forest walking (*n* = 24) (Con.) Urban walking (*n* = 24)	Adult (Male)	5 min after intervention	BP HRV	[LF/HF **] Exp. (1.5) was lower than Con. (1.9) [SBP] Exp. (114) and Con. (116) were not statistically different [HF **] Exp. (4.4) was higher than Con. (3.8)	NM	⨁⨁⨁⨁ High
Psychological health indicators	[10] Bang (2016)	O	Healthy person	(Exp.) Urban Forest walking (*n* = 18) (Con.) No intervention (*n* = 27)	Adults (Male and female)	Promised time	Depression (BDI) QOL	[Depression] Exp. (5.11) and Con. (6.44) were not statistically different [QOL *] Exp. (23.94) was higher than Con. (20.70)	NM	⨁⨁◯◯ Low
	[11] Brown (2014)	X	Healthy person	(Exp1.) Urban forest walking (*n* = 27) (Exp2.) Urban street walking (*n* = 27) (Con.) No intervention (*n* = 19)	Adults (Male and female)	On-site (physiological data), online (psychological data) Following the 8 week intervention	SF-8 general health SF-8 physical health SF-8 mental health	[SF-8_general health] Exp. 1 (50.2), Exp. 2 (50.5), and Con. (47.8) were not statistically different [SF-8_physical health] Exp. 1 (54.9), Exp. 2 (51.8), and Con. (53.4) were not statistically different [SF-8_mental health] Exp. 1 (53.0), Exp. 2 (50.1), and Con. (47.4) were not statistically different	NM	⨁⨁⨁◯ Moderate
	[21] Chun (2017)	O	Patient (Chronic stroke)	(Exp.) meditation walking and 5 sense experience in forest (*n* = 30) (Con.) meditation and walking in urban hotel (*n* = 29)	Adults and elderly (36~79 years) (Male and female)	Immediately before and after programs	Depression (BDI) Depression (HAM-D17) Anxiety (STAI)	[BDI ***] Exp. was decreased (pre 14.2 vs. post 1.2) [BDI] Con. was not statistically different (pre 14.3 vs. post 14.0). [HAM-D17 ***] Exp. was decreased (pre 7.1 vs. post 1.6) [HAM-D17] Con. was not statistically different (pre 7.2 vs. post 7.1) [STAI ***] Exp. was decreased (pre 38.1 vs. post 27.6). [STAI ***] Con. was increased (pre 34.3 vs. post 44.4)	NM	⨁⨁⨁◯ Moderate
	[22] Huber (2019)	X	Patient (CLBP)	(Exp 1) Green exercise (*n* = 27) (Exp 2) Green exercise and balneotherapy (*n* = 26) (Con) No intervention (*n* = 27)	Adults (19~65 years) (Male and female)	Day 1, after the intervention (day 8), after 4 months (day 120)	QOL (SF-36 total) QOL (SF-36 physical) QOL (SF-36 mental) QOL (WHO-5)	Short-term effect (day 8) Exp. 1 was not changed. Exp. 2 was increased SF-36 total *, physical *, and WHO-5 ** ^†^ Long-term effect (day 120) All indices are not significant. ^†^	NM	⨁⨁⨁◯ Moderate
	[15] Jia (2016)	X	Patient (COPD)	(Exp.) Forest walking (*n* = 10) (Con.) Urban walking (*n* = 8)	Elderly (Male and female)	Before breakfast the day after intervention	Mood (POMS)	[Mood-T] Exp. was lower than Con.* ^†^ [Mood-D, Mood-A, Mood-V, Mood-F, and Mood-C] Exp. and Con. were not statistically different ^†^	NM	⨁⨁◯◯ Low
	[41] Mao (2017)	X	Patient (CHF)	(Exp.) Forest walking (*n* = 23) (Con.) City walking (*n* = 10)	Elderly (65~85 years) (Male and female)	5~10 min after intervention	Mood (POMS)	[Mood-T *, Mood-D *, Mood-A *, and Mood-C *] Exp. was lower than Con. ^†^	NM	⨁⨁◯◯ Low
	[14] Mao(a) (2012)	X	Patient (HTN)	(Exp.) Forest walking (*n* = 12) (Con.) City walking (*n* = 12)	Adult	30~40 min after intervention (physiological data), next evening (psychological data)	Mood (POMS)	[Mood-T, Mood-V] Exp. and Con. Were not statistically different ^†^ [Mood-D *, Mood-A *, Mood-F *, and Mood-C *] Exp. was lower than Con. ^†^	NM	⨁⨁◯◯ Low
	[23] Mao(b) (2012)	X	Healthy person	(Exp.) Forest walking (*n* = 10) (Con.) City walking (*n* = 10)	Adult (Male)	Before breakfast the day after intervention	Mood (POMS)	[Mood-T *, Mood-D *, Mood-A *, and Mood-F *] Exp. was lower than Con. ^†^ [Mood-V *] Exp. was higher than Con. ^†^	NM	⨁⨁◯◯ Low
	[19] Shin (2012)	X	Patient (Alcoholic)	(Exp.) Forest camping (*n* = 47) (Con.) Normal daily routine (*n* = 45)	Adult	End day of the final session of the camp	Depression (BDI)	[Depression] Exp. (5.52) and Con. (15.36) were not statistically different	NM	⨁⨁◯◯ Low
	[20] Song (2019)	O	Healthy person	(Exp.) forest walking (*n* = 72) (Con.) city walking (*n* = 72)	Adult (Young female)	5 min after intervention	Anxiety (STAI) Mood (POMS)	[Total Mood **] Exp. (0.1) was lower than Con. (7.7) [STAI **] Exp. (34.8) was lower than Con. (45.3).	NM	⨁⨁◯◯ Low
	[13] Wu (2020)	O	Patient (HTN)	(Exp.) Sit in forest (*n* = 20) (Con.) Sit in suburban (*n* = 11)	Elderly	After intervention	Mood (POMS)	[Mood-T *] Exp. (12.90) was lower than Con. (15.55) [Mood-D *] Exp. (25.72) was lower than Con. (29.82) [Mood-A] Exp. and Con. were not significantly different ^†^ [Mood-V *] Exp. (26.90) was higher than Con. (24.36) [Mood-F *] Exp. (13.80) was lower than Con. (15.55) [Mood-C *] Exp. (13.75) was lower than Con. (16.64)	NM	⨁⨁⨁◯ Moderate
	[40] Lee (2014)	X	Healthy person	(Exp) Forest walking (*n* = 24) (Con) Urban walking (*n* = 24)	Adult (Male)	5 min after intervention	Anxiety (STAI) Mood (POMS)	[Mood-T **] Exp. (35.6) was lower than Con. (41.6) [Mood-A **] Exp. (37.7) was lower than Con. (39.0) [Mood-F **] Exp. (36.1) was lower than Con. (41.4) [Mood-C **] Exp. (42.2) was lower than Con. (44.3) [Mood-V **] Exp was higher than Con. ^†^	NM	⨁⨁⨁⨁ High

Note: ^†^ = No statistics; NM = Not mentioned; NS = Not significant; NA = Not available; BP = Blood Pressure; HR = Heart Rate; CAD = Coronary Artery Disease; COPD = Chronic Obstructive Pulmonary Disease; CLBP = Chronic Low Back Pain; CHF = Chronic Heart Failure; HTN = Hypertension; LF = Low Frequency (marker of cardiac parasympathetic control); HF = High Frequency (marker of cardiac sympathetic control); LF/HF = Low Frequency/High Frequency (index of sympathetic to parasympathetic autonomic activity); SPO2 = Saturation of Percutaneous Oxygen; BDI = Beck Depression Inventory; QOL = Quality of Life; SF-8 = 8-item Short Form health survey; HAM-D17 = 17-item version of the Hamilton Depression rating scale; STAI = Spielberger State-Trait Anxiety Inventory; SF-36 = 36-item Short Form health survey; WHO-5 = World Health Organization wellbeing index; POMS = Profile of Mood State, Mood-T = Mood-Tension anxiety, Mood-D = Mood-Depression dejection, Mood-A = Mood-Anger hostility, Mood-V = Mood-Vigor activity, Mood-F = Mood-Fatigue inertia, Mood-C = Mood-Confusion bewilderment. ⨁⨁⨁⨁ = high, ⨁⨁⨁◯ = moderate, ⨁⨁◯◯ = low, O = included, X = excluded, * = *p* < 0.05, ** = *p* < 0.01, *** = *p* < 0.001

**Table 2 ijerph-19-10512-t002:** Program characteristics of included studies.

First Author (Year Published)	Season or Weather	Session and Duration	Caution before or during Intervention	Homogeneous Environment Setting	Program Type
[10] Bang (2016)	September~ November	40 min × 2 times per week × 5 weeks Multi-session	NM	NM	(Exp.) During lunchtime walking Park or palace near workplace
[11] Brown (2014)	April~ July	20 min × 2 times per week × 8 weeks Multi-session	NM	NM	(Exp.) During lunchtime walking route approximately 2 km in length
[36] Calogiuri (2016)	September	45 min × 1 time per day × 2 consecutive days Biking 25 min + rubber band session 20 min	Restrict intake of coffee and nicotine, avoid any other physical activity	Control subjects did not have visual contact with nature.	(Exp.) Nature exercise (biking and rubber band exercise) (Con.) Indoor exercise (biking & rubber band exercise)
[21] Chun (2017)	NM	4 day and 3 night Multi-session	NM	Same duration and activities	Meditation and walking (Both group), five senses experience (Exp.) (Exp.) Staying at a recreational forest site (Con.) Staying in an urban hotel
[37] Grazuleviciene (2016)	May~ September	30 min per day × 7 consecutive days Multi-session	Refrain from consuming caffeine or food for at least 60 min prior to measurement	Same walking speed controlled by a trained nurse to reach the personal exercise capacity	Seven day field experiment (Exp.) 30 min single walking
[22] Huber (2019)	September ~January	5 h per day × 6 days Multi-session	NM	Hosted in comparable hotels and receiving the same meals	Hiking tours in the mountains (Exp. 1, Exp. 2), additional balneotherapy for 20 min at 37 °C (Exp. 2) (Exp. 1) Green exercise (Exp. 2) Green exercise + balneotherapy (Con.) Control
[15] Jia (2016)	August	90 min × 2 times per day × 3 consecutive days	Not mentioned	Same hotel	Walking during 90 min each in morning and afternoon
[39] Lee (2011)	NM	15 min × 1 time per day × 1 day One session	No smoking No alcohol No caffeine	Stay in the same hotel before intervention	Three day field experiment Group (Exp. vs. Con.) switched (Exp.) Viewing
[40] Lee (2014)	August ~September	12 min × 1 time per day × 1 day One session	No physical activity No smoking No alcohol	Course length & Flat roadside	Two day field experiment Group (Exp. vs. Con.) switched (Exp.) Walking in four different forest area
[14] Mao(a) (2012)	September	1.5 h × 2 times per day × 7 days Multi session	Avoiding strenuous exercise and any stimulation activities before sleeping	Predetermined course, Stay in the sample hotel before intervention	(Exp.) Walking during 90 min each in morning and afternoon per day for 7 days
[23] Mao(b) (2012)	September	1.5 h × 2 times per day × 1 day One session	Avoiding strenuous exercise and any stimulation activities before sleeping	Similar condition hotel	(Exp.) walked unhurried pace for about 1.5 h with a 10 min rest during the walk before noon and afternoon in the forest (Con.) walked with same procedure in the city
[41] Mao (2017)	August	1.5 h × 2 times per day × 4 days Multi session	No smoking No caffeinated beverages Controlled of physical activity and all foods	Similar distance (5–10 min walk) from the site hotel Same flat path	Four day forest bathing (Exp.) Walking
[12] Niedermerier (2017)	NM	3 h One session	NM	NM	(Exp.1) hiked uphill for 6 km on single trails and forest roads to a mountain hut (1500 m) with a view of the mountainous region. (Uphill 90 min, resting 10 min, downhill 70 min) (Exp.2) treadmill walking (uphill 90 min, resting 10 min, level walking on the same treadmills (70 min) (Con.) Quiet room with computer.
[17] Shin (2012)	Summer	9 day forest healing camp	NM	NM	(Exp.) First 3 day program (Nature game, nature interpretation etc.), second 3 day program (mountain climbing, tracking, orienteering, etc.), last 3 day program (nature meditation, counseling in forest environment, etc.) (Con.) Normal daily routine
[20] Song (2019)	August ~September	15 min × 1 time (Approximately 1 km)	No alcohol No tobacco No caffeine	NM	(Exp.) Walking at 6 forest sites (Con.) Walking at 6 city sites Groups switched field site on next day. Subjects walk on different days for each of the 6 sites
[13] Wu (2020)	October	2 h × 1 time per day × 1 day	No activity No alcohol No tobacco No caffeine drink	Similar condition hotel	(Exp.) First day after lunch sit quietly each place. Second day, sit quietly on the morning and afternoon each place
[38] Zeng (2020)	September	15 min landscape and 15 min walking × 1 time per day × 3 days	No strenuous exercise No stimulation activity before sleeping	Same distance (300 m) from the site hotel Similar meal	Three day bamboo forest therapy (Exp.) Viewing and walking

Note: NM = Not mentioned; NS = Not significant; NA = Not available; BP = Blood Pressure; HR = Heart Rate; CAD = Coronary Artery Disease; COPD = Chronic Obstructive Pulmonary Disease; CLBP = Chronic Low Back Pain; CHF = Chronic Heart Failure; HTN = Hypertension; LF = Low Frequency (marker of cardiac parasympathetic control); HF = High Frequency (marker of cardiac sympathetic control); LF/HF = Low Frequency/High Frequency (index of sympathetic to parasympathetic autonomic activity); SPO2 = Saturation of Percutaneous Oxygen; BDI = Beck Depression Inventory; QOL = Quality of Life; SF-8 = 8-item Short Form health survey; HAM-D17 = 17-item version of the Hamilton Depression rating scale; STAI = Spielberger State-Trait Anxiety Inventory; SF-36 = 36-item Short Form health survey; WHO-5 = World Health Organization wellbeing index; POMS = Profile of Mood State.

## Data Availability

The authors confirm that the data supporting the findings and conclusion for this study are available within the article and its Appendix A.

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
