# Peer review of "Does Forest Therapy Have Physio-Psychological Benefits? A Systematic Review and Meta-Analysis of Randomized Controlled Trials"

_ijerph, 2022, doi:10.3390/ijerph191710512_

Round 1

Reviewer 1 Report

Unfortunately, not too much of a change was made as compared to previous versions of the submission. Thus, structural problems remain. The Authors have already received the necessary suggestions to do the job.

Author Response

Thank you for your opinion. As you pointed out, the number of papers included in the study is small to comprise a sub-factor of a meta-analysis. Therefore, we have changed the main content of the paper into a systematic review, and meta-analysis is included ancillarily. Please re-evaluate the study on the basis of a systematic review.

Reviewer 2 Report

Comments to the author

This article analyzes the effects of forest therapy from a number of papers. However, the points raised have not yet been remedied.

1.

Line 461~

“Since we confirmed that forest therapy has a positive effect on depression,~.”

This sentence is abrupt and gives the impression that forest therapy has had a positive effect on disease of depression ( or depressed patients). The fact that improvement was observed in subjective symptoms such as POMS in healthy subjects means improvement in depressive statement, so words such as “depression status” should be used. There are other places where the use of the word “depression status” would be more appropriate.

In addition, the following statements are included.

Line372~

“As a result of the meta-analysis, forest therapy was found to be effective in reducing depression in adults (SMD=-1.46, CI: -2.80-0.12). Kotera et al. also confirmed that forest 373 therapy was effective in reducing depression [47]. “

However, Kotera's paper states that “Shinrin-yoku was deemed to have a greater effect on anxiety, than depression and anger, ~ “ and is not appropriate as a reference to cite when describing improvement in depression. 

2.

The authors cited a definition of forest therapy from Reference 6, which was written in Korean. Then the definition of forest therapy is a definition that is accepted only among those who can read Korean and may differ from the world's perception. At the very least, papers published in English should be adopted. This is a major problem with the underlying definition.

3.

The papers selected for this study include a variety of activities in the forest, and some of the papers include exercises that are obvious. For example, reference 36 states "exercise sessions, each including a biking bout and a circuit-strength sequence using elastic rubber bands”. Therefore, the authors should mention this in the LIMITATION because it includes papers that may have different effects depending on the intensity of exercise, i.e., they may be looking at exercise effects.

Author Response

Attached file. 

This manuscript is a resubmission of an earlier submission. The following is a list of the peer review reports and author responses from that submission.

Round 1

Reviewer 1 Report

I acknowledge that there was a certain improvement of the work. However, in my opinion its quality does not meet the standards of the journal yet.

The manuscript remains weakly constructed and organized. The authors should do an extensive job trying to deconstruct and reconstruct it. In particular, the first part of the Discussion sounds like a repetition of the results of the systematic review.

These considerations apply not only to content but also to form. Indeed, English still needs improvement: some sentences are unclear and, overall, the quality and fluency of the manuscript are poor.

Also, the Authors should pay more attention to details. For instance, the Abstract does not reflect the changes that have been made in the main text, forest plots remained of the same (low) quality, and references are not in numerical order.

Reviewer 2 Report

Comments to the author

This is an article that attempts a meta-analysis analysis of the effects of forest bathing. However, before the analysis, I think it is necessary to review the way the references were adopted.

1.

In investigating the effects of Forest Bathing (Forest Bathing), Tsunetsugu et al. describe it as " We have conducted physiological experiments both in actual forests and in the laboratory in order to elucidate the physiological effects of the total forest environment or certain elements of the forest environment, such as the odor of wood, the sound of running stream water, and the scenery of the forest." (Yuko Tsunetsugu, Bum-Jin Park, Yoshifumi Miyazaki. Environ Health Prev Med. 2010 Jan;15(1):27-37.)

In addition, Hansen et al. described that “〜the human physiological and psychological systems associated with the practice of Shinrin-Yoku (SY), also known as Forest Bathing FB (FB). SY is a traditional Japanese practice of immersing oneself in nature by mindfully using all five senses. “ (Hansen MM, Jones R, Tocchini K. Int J Environ Res Public Health. 2017 Jul 28;14(8):851.)

In other words, forest bathing began as an investigation with the goal of seeing how the human body and mind would respond by staying in a forest environment, and the study came to be known as forest medicine.

In reviewing the effects of forest bathing, then, the authors should adopt articles that investigate the results of spending time quietly in the forest, walking, and using the five senses. Papers on other types of exercise should not be juxtaposed as "results of forest bathing" because of the bias of exercise effects.

Exercise itself has been reported to increase blood pressure, improve vitality, and more recently to improve depression through the secretion of hormones such as IGF-1.

If authors were to adopt the articles on exercise, they should consider it as "the effect of combining forest bathing with exercise" in a separate category from the above conditions.

2.

Line 128: Bias occurs when only articles written in Korean are adopted. In that case, papers written in each country's specific language should also be adopted. In this case, I think it should be limited to papers written in English only.